# How Changes in ABA Accumulation and Signaling Influence Tomato Drought Responses and Reproductive Development

**Paolo Korwin Krukowski** [ID], **Sara Colanero** [ID], **Aldo Sutti** [ID], **Damiano Martignago** [ID] and **Lucio Conti** *[ID]

Department of Biosciences, University of Milan, 26 Celoria Street, 20133 Milan, Italy
* Correspondence: lucio.conti@unimi.it

**Abstract:** Water deficit conditions trigger the production of a chemical signal, the phytohormone abscisic acid (ABA), which coordinates multiple responses at different temporal and spatial scales. Despite the complexity of natural drought conditions, the modulation of ABA signaling could be harnessed to ameliorate the drought performances of crops in the face of increasingly challenging climate conditions. Based on recent studies, increasing ABA sensitivity can lead to genotypes with improved drought resistance traits, with sustained biomass production in water-limiting environments and little or no costs with respect to biomass production under optimal conditions. However, variations in ABA production and sensitivity lead to changes in various aspects of reproductive development, including flowering time. Here we provide an updated summary of the literature on ABA-related genes in tomato and discuss how their manipulation can impact water-deficit-related responses and/or other developmental traits. We suggest that a better understanding of specific ABA components' function or their expression may offer novel tools to specifically engineer drought resistance without affecting developmental traits.

**Keywords:** *Solanum lycopersicum*; tomato; abscisic acid; ABA; drought stress; reproductive development; ABA biosynthesis; ABA signaling

## 1. Introduction

Ongoing climatic changes and the rapid depletion of water resources generate uncertainties about crop productivity [1]. Limiting water use while maintaining high levels of productivity is still a formidable challenge for agriculture and the plant science community [2,3]. Part of the challenge depends on the complexity of meteorological drought scenarios which may influence plants at different developmental stages, with variable duration and intensities [4]. On the other hand, plants can adopt multiple strategies to cope with drought stress conditions, including drought escape, avoidance, and tolerance, which may be constitutive or plastic—i.e., influenced by complex interactions between genotypes and water deficit signals [5]. To escape from water-deficit stress, plants complete their life cycle before a drought becomes too severe. Tolerance to drought is acquired by osmotic adjustment, reactive oxygen species (ROS) scavenging, and the activation of stress-related genes. Plants can also avoid dehydration during transient periods of water deficiency by stomatal closure and shoot growth suppression—both leading to reduced transpiration and increased root growth to explore water sources at greater depths. The number and aperture of stomatal pores are finely balanced to maintain an optimal lifetime accumulation of biomass (through carbon fixation) according to the available water in the soil. Because of the trade-off between transpiration and growth, improvements to drought tolerance traits through reductions in water loss frequently impair biomass accumulation.

A key goal for breeders is to generate genotypes with sustained biomass production in water-limiting environments. From a physiological perspective, water loss is largely due to stomatal transpiration, regulated through stomatal closure based on soil water availability. Drought promotes a reduction in stomatal conductance and a consequent

decrease in photosynthetic rates and carbon assimilation. To overcome these limitations, plants also modify their osmotic balance to improve both water absorption and retention through the production of osmoprotective compounds, such as proline, while counteracting the ROS-triggered oxidative damage by producing ROS-scavenging enzymes [6]. To achieve drought resistance, plants need to maintain a balance between stress tolerance and avoidance mechanisms to maintain growth and reach reproductive development, which is often altered in response to drought conditions.

The literature offers many possible molecular targets amenable to genetic manipulation that could improve drought tolerance traits [7]. A common thread of the manifold responses to water deficits is the production of the phytohormone abscisic acid (ABA). ABA regulates stomatal activity and water homeostasis by directly affecting plant water status and water-use efficiency (WUE), i.e., the ratio of biomass produced per unit of water consumed. Shoot growth inhibition and reduced productivity during a water deficiency are the result of the rapid accumulation of ABA, leading to stomatal closure and reduced $CO_2$ assimilation [8]. In roots, ABA promotes growth, and the increased roots biomass enables plants to explore the soil volume and improve water uptake [9,10].

Enhancing ABA responses could be a promising approach to increase drought resistance. Still, it may also lead to growth penalties under the prevailing agricultural conditions. Recent reports indicate that it is possible to increase ABA sensitivity in crops to improve drought resistance without growth reductions (i.e., no costs in carbon fixation and biomass production) [11–13]. Interestingly, variations in ABA sensitivity also influence flowering time [11,14,15], fruit development, and seed germinability [6]. Thus, ABA has multiple roles in plant development, independent of water deficits, including the regulation of primary metabolism [16]. Here, by focusing on tomato (*Solanum lycopersicum*), a crop plant that will be subject to the adverse effects of climate change in the field, we aim to provide a comprehensive summary of the available mutants and transgenic resources to engineer ABA biosynthesis and signaling. We discuss how these mutations can impact drought tolerance, while also considering their possible effects on floral transition and fruit development.

## 2. Genetic Modulation of ABA Biosynthesis and Signaling in Tomato

ABA signaling and biosynthesis have been studied in tomato through different approaches in at least six different cultivars (cv.), with microtom and Alisa Craig being the most used. Historically, much of our knowledge on ABA biosynthesis in tomato is derived from forward genetics screens (i.e., from phenotype to genotype), whereas signaling mechanisms are studied mostly through reverse genetics (i.e., from genotype to phenotype) using gene silencing or overexpression approaches. We provide a table summarising the mutant and transgenic lines discussed herein (Table 1), although direct comparisons of the described phenotypes are not always straightforward due to background effects or a lack of standardised experimental conditions.

**Table 1.** Tomato ABA biosynthesis and signaling-related mutants and transgenic lines. The table summarizes mutants and transgenic lines described in the review. For each manipulated gene, we provide the following: gene name, Sol Genomics locus ID, name of the mutant or transgenic line (according to references), type of modification (i.e., loss of function, overexpressor, RNAi), discussed phenotypes, line cultivar, genetic manipulation technique, and references.

| Gene Name | Sol Genomics Locus Id | Mutant of Transgenic Line | Type of Genetic Modification | Phenotypes | Cultivar | Genetic Manipulation Technique | References |
|---|---|---|---|---|---|---|---|
| **Biosynthesis** | | | | | | | |
| *SlZEP* | solyc02g090890 | *hp3* | loss of function | • Carotenoid decrease in leaves, flowers and fruits <br> • Reduction in ABA content <br> • Lower stomatal closure <br> • Higher stomatal conductance <br> • Hampered growth in field but not in optimal drought conditions <br> • More chloroplasts in green fruits | M82 | EMS-mediated mutagenesis | [17–19] |
| *NSY* | solyc12g041880 | *nxd1-1* | loss of function | • No changes in leaf ABA levels <br> • Slow wilting in drought <br> • In greenhouse conditions, tolerates stress better than WT <br> • No changes in yield and biomass in field | M82 | EMS-mediated mutagenesis | [18–20] |
| | | *ndx1-2* | loss of function | | M82 | EMS-mediated mutagenesis | |
| *SlNCED1* | solyc07g056570 | *not* | loss of function | • Fast-wilting <br> • High transpiration <br> • Abnormal stomatal size <br> • Decreased stomatal function <br> • Low ABA content in leaves and fruits | Alisa Craig | X-ray-mediated mutagenesis | [21–24] |
| | | D9 SP5/SP6 | overexpressor | • Low transpiration (normal conditions), high ABA content in leaves | Alisa Craig | Agrobaterium transformation | [25] |
| *FLACCA* | solyc07g066480 | *flc* | loss of function | • Fast-wilting <br> • Hampered vegetative growth <br> • Drying leaves show high transpiration <br> • Dense and abnormal stomatal size with reduced modulation <br> • Low ABA content in leaves and fruits <br> • Fails to accumulate ABA following water deprivation <br> • Smaller fruits | Rheinlands Ruhm | X-ray- mediated mutagenesis | [23,26–29] |
| *SITIENS* | solyc01g009230 | sit | loss of function | • High transpiration (normal conditions) | Rheinlands Ruhm | X-ray- mediated mutagenesis | [24,26,28,30] |
| **Signaling** | | | | | | | |
| *SlPYL9* | solyc09g015380 | SlPYL9-OE | overexpressor | • Low leaf water loss during drought <br> • Faster fruit ripening <br> • Reduced fruit shelf-life | Microtom | Agrobacterium transformation | [31,32] |
| | | SlPYL9-RNAic | RNAi | • High water loss during drought <br> • Slow fruit ripening <br> • Increased fruit shelf life | | | |
| *SlPP2C49* | solyc06g076400 | SlPP2C49-OE | overexpressor | • High leaf water loss during drought <br> • Slow fruit ripening | Microtom | Agrobacterium transformation | [33] |
| | | *Sl*PP2C49-RNAi | RNAi | • Low leaf water loss during drought <br> • Fast fruit ripening | | | |

**Table 1.** *Cont.*

| Gene Name | Sol Genomics Locus Id | Mutant of Transgenic Line | Type of Genetic Modification | Phenotypes | Cultivar | Genetic Manipulation Technique | References |
|---|---|---|---|---|---|---|---|
| *SlPP2C30* | solyc03g121880 | *Sl*PP2C30-OE | overexpressor | • High leaf water loss during drought<br>• Low ABA content in leaves during drought<br>• Slow fruit ripening<br>• Reduced growth | Microtom | Agrobacterium transformation | [34] |
| | | *Sl*PP2C30-RNAi | RNAi | • Low leaf water loss during drought<br>• High ABA content in leaves during drought<br>• Faster fruit ripening | | | |
| *SlOST1 (Snrk2.3)* | solyc01g108280 | *Slost1-1* | loss of function | • Generally drought-sensitive<br>• Late flowering in both LD and SD<br>• Late flowering in drought and normal conditions<br>• Higher stomatal conductance | Microtom | CRISPR deletion | [15] |
| | | *Slost1-2* | loss of function | | | | |
| | | *Slost1-3* | loss of functiont | • Generally drought- sensitive<br>• Late flowering in LD and SD<br>• Late flowering in drought and normal conditions<br>• Higher stomatal conductance | Alisa Craig | | |
| | | *Slost1-4* | loss of function | | | | |
| *SlAREB1* | solyc04g078840 | LA1 | overexpressor | • Lower water loss in prolonged drought | CL5915-93D4-1-0-3 | Agrobacterium transformation | [35] |
| | | LA2 | | • Lower water loss in prolonged drought | | | |
| | | LA3 | | • Lower water loss in prolonged drought | | | |
| | | S1 | overexpressor | • During droughts, there is an increased survival rate due to low leaf water loss and better photosynthetic efficiency and ROS scavenging | Moneymaker | Agrobacterium transformation | [36] |
| | | S2 | | | | | |
| | | S3 | | | | | |
| | | S6 | | | | | |
| | | A1 | RNAi | • During droughts, there is a decreased survival rate due to high leaf water loss and worsening photosynthetic efficiency and ROS scavenging | | | |
| | | A3 | | | | | |
| | | A4 | | | | | |
| | | A6 | | | | | |
| *SlNAC6* | solyc10g055760 | R-#6 | RNAi | • Lower ABA content in normal conditions<br>• Lower stomata sensitivity to ABA and osmotic stress<br>• Less tolerant to osmotic stress<br>• Dwarf | Microtom | Agrobacterium transformation | [37] |
| | | R-#15 | | | | | |
| | | R-#18 | | | | | |
| | | OE-#2 | overexpressor | • Higher ABA content in normal conditions<br>• Higher stomata sensitivity to ABA and osmotic stress<br>• More tolerant to osmotic stress probably due to lower stomata water loss<br>• Late flowering in normal conditions | | | |
| | | OE-#3 | | | | | |
| | | OE-#11 | | | | | |
| *SlGRAS4* | solyc01g100200 | RNAi#10 | RNAi | • Impaired ROS scavenging in drought stress | Microtom | Agrobacterium transformation | [38] |
| | | RNAi#15 | | | | | |
| | | RNAi#16 | | | | | |

**Table 1.** *Cont.*

| Gene Name | Sol Genomics Locus Id | Mutant of Transgenic Line | Type of Genetic Modification | Phenotypes | Cultivar | Genetic Manipulation Technique | References |
|---|---|---|---|---|---|---|---|
| | | OE#12 | | | | | |
| | | OE#18 | overexpressor | • Enhanced ROS scavenging in drought stress | | | |
| | | OE#27 | | | | | |
| *SlMYB50* | solyc06g071690 | RNAi01 | RNAi | • Overall better drought tolerance<br>• Low leaf water loss<br>• High proline accumulation<br>• High photosynthetic efficiency in drought<br>• Enhanced ROS scavenging | Alisa Craig | Agrobacterium transformation | [39] |
| | | RNAi02 | | | | | |
| | | RNAi04 | | | | | |
| *SlMYB55* | solyc10g044680 | RNAi06 | RNAi | • Overall better drought tolerance<br>• Low leaf water loss<br>• High proline accumulation<br>• High photosynthetic efficiency in droughts<br>• Enhanced ROS scavenging<br>• Early flowering<br>• Increased number of flowers | Alisa Craig | Agrobacterium transformation | [40] |
| | | RNAi08 | | | | | |
| | | RNAi09 | | | | | |
| *SlHB2* | solyc05g006980 | RNAi1 | RNAi | • Low water loss<br>• High proline accumulation<br>• Enhanced ROS scavenging | Alisa Craig | Agrobacterium transformation | [41] |
| | | RNAi2 | | | | | |
| | | RNAI3 | | | | | |
| | | RNAi4 | | | | | |
| | | RNAi5 | | | | | |
| | | RNAi6 | | | | | |
| *SlbHLH96* | solyc11g056650 | OE-*Sl*bHLH96-2 | overexpressor | • High leaf ABA content<br>• Exaggerated drought-induced stomatal closure<br>• Enhanced ROS scavenging | Alisa Craig | Agrobaterium transformation | [42] |
| | | OE-*Sl*bHLH96-17 | | | | | |

### 2.1. Tomato ABA Biosynthetic Mutants and Transgenic Plants

In flowering plants, ABA is synthesized via the carotenoid pathway by the cleavage of β-carotene metabolites called xanthophylls, sharing the same intermediate molecular pool of gibberellic acid (GAs). A summary of the steps of ABA's biosynthesis in tomato is provided in Figure 1 and we invite readers to refer to specialised and comprehensive reviews on the topic (e.g., [43]). The first steps of ABA biosynthesis take place in the plastid, with the oxidative cleavage of zeaxanthin into all-trans-violaxanthin by the enzyme zeaxanthin epoxidase (ZEP). The ZEP activity is encoded by the *HIGH PIGMENT 3* (*HP3*) locus in tomato. The *hp3* mutants are isolated from an ethyl methanesulfonate (EMS) mutagenesis screen (cv. M82) [17,18]. The identified recessive mutation causes an amino acid substitution (K142E) in the ZEP protein sequence, leading to a reduced enzymatic function and the conversion of zeaxanthin to violaxanthin. The *hp3* mutants are characterized by increased levels of carotenoids in leaves and red fruits, whereas flowers present a light yellowish color due to the decreased accumulation of violaxanthin and neoxanthin. The observed increase in the carotenoids in the fruits of *hp3* mutants (and other ABA biosynthetic mutants) is attributed to an increase (approx. 30%) in plastids in the cells isolated from mature green fruits. Under field conditions, *hp3* mutant plants synthetize almost 68% less ABA compared to wild type (WT) plants, which translates into an increased transpiration due to a reduced stomatal closure [17,19]. Consequently, mutants grown under stressful conditions tend to wilt much faster than the WT, displaying reduced biomass and up to a 60% lower yield. However, none of these defects could be observed when well-watered and grown under optimal conditions.

The ZEP product trans-violaxanthin can be further processed by two different enzymes: xanthophyll 9-cis isomerase (XISO) which converts it into 9-cis violaxanthin; or neoxanthin synthase (NSY), which produces trans-neoxanthin. Both products can be used as a substrate for 9-cis-epoxycarotenoid dioxygenase (NCED) [43]. The enzymatic activity responsible for the conversion of violaxanthin to neoxanthin remained unknown up until a few years ago. Neuman et al. [19] isolated two neoxanthin-deficient mutants (*ndx1-1* and *ndx1-2*) which lack neoxanthin due to a mutation in the NSY gene (*Solyc12g041880*) located on chr 12 [18], which corresponds to Arabidopsis ABA-deficient 4 (*ABA4*) [20]. In *ndx1-1*, a C to T substitution creates a new splicing site leading to an early stop codon, while in *ndx1-2* an A to T substitution in the second intron causes a decrease in its transcript levels. In field trials, no significant reductions in biomass and yield were observed in *ndx1-1* and *ndx1-2* mutants. Under greenhouse conditions, the leaf water loss of these two mutants under a water deficiency was comparable to the WT, yet they could better tolerate a prolonged water deficit. Interestingly, despite their increased levels of drought tolerance (or avoidance) compared with WT plants (17 days for WT and 21 days for *ndx1-1*), the *ndx1* plants did not show significant differences in ABA accumulation compared with the WT plants [19]. This suggests that, in tomato, NSY function has a minor contribution to ABA accumulation in leaves, although its function might still be critical in specific aspects of its water deficit response.

In the downstream steps of ABA biosynthesis, all-trans-viola/neoxanthin intermediates are further modified into xanthoxin by NCEDs, regarded as a rate-limiting step in ABA biosynthesis. In Arabidopsis, nine NCED-encoding genes have been identified in the complete genome sequence [44], whereas in tomato this activity is encoded by two highly related genes, *SlNCED1* and *2* [45]. From xanthoxin, bioactive ABA is synthesized in two steps. First, short-chain dehydrogenase/reductase 1 converts xanthoxin into abscisic aldehyde. Secondly, abscisic aldehyde oxidase apoenzyme encoded by the *SITIENS* (*SIT*) gene in cooperation with a molybdenum cofactor encoded by the *FLACCA* (*FLC*) gene [28] produce an ABA active molecule from its aldehyde.

*Notabilis* (*not*) (cv. Alisa Craig) is a null mutant characterized by a frameshift in the *SlNCED1* gene [21]. The loss of *SlNCED1* function leads to fast-wilting plants with reduced ABA levels in their leaves. The growth of *not* mutants is generally hampered by their fast wilting, even when well-watered [22], although no comparable differences

were observed between *not* and WT plants under high-humidity conditions [23]. The leaves of *not* mutants present abnormally large stomata which constitutively open as a result of insufficient levels of ABA accumulation [23,24]. Virus-induced gene silencing (VIGS) of *SlNCED1* causes a significant delay in fruit ripening, a trait not assessed in *not* mutants [23,46]. Three stable *SlNCED1* overexpressing lines—named D9 (using a tandem cauliflower mosaic virus—*CaMV 35S* promoters), SP5, and SP6 (using chimeric promoters) (cv. Alisa Craig)—were characterized by an over-guttation phenotype (i.e., an exaggerated water loss from hydathodes) possibly caused by highly reduced levels of transpiration. D9 and SP5 transgenic plants further showed constitutively high ABA levels in their leaves, increased seed dormancy, and generally lower stomatal conductance [25].

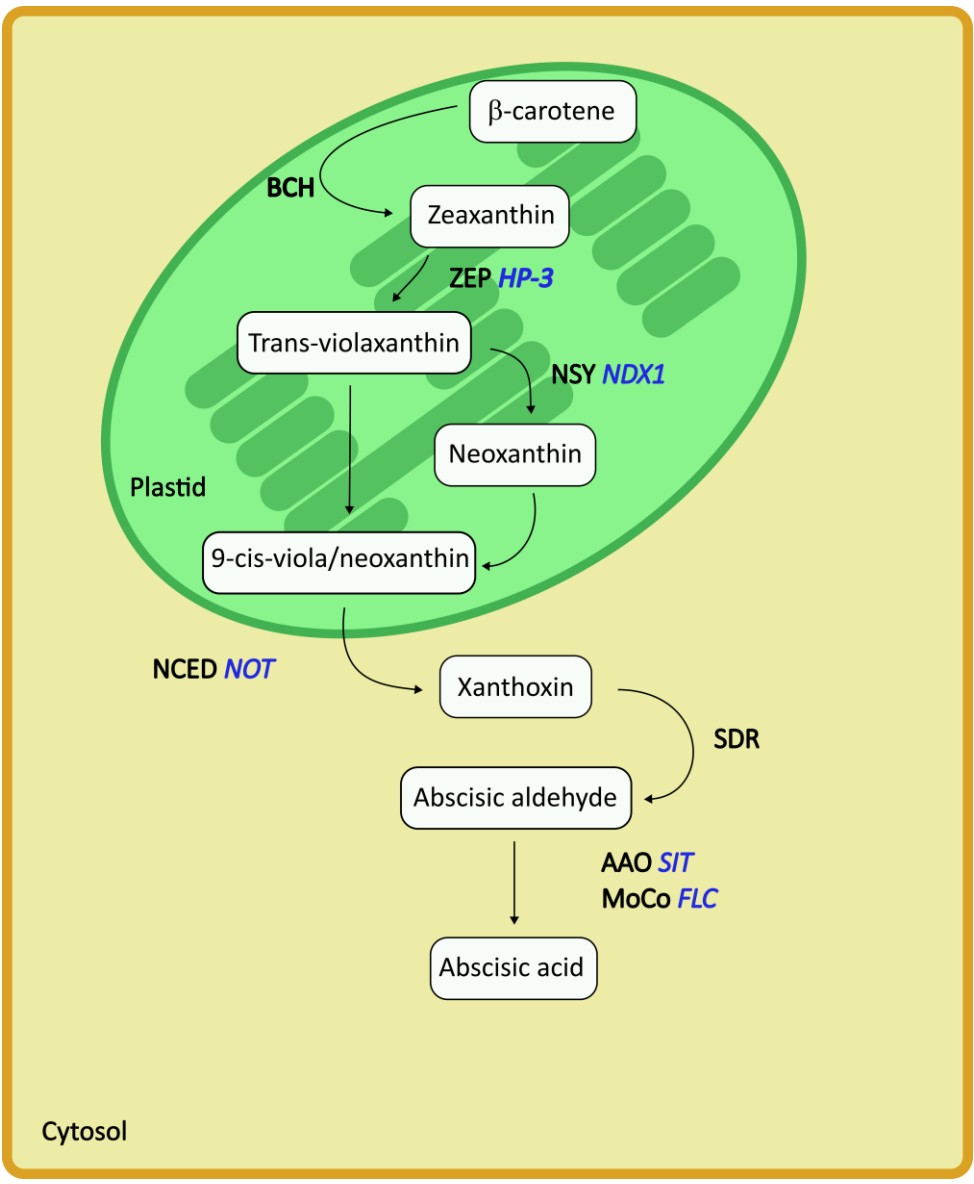

**Figure 1.** ABA biosynthesis in tomato. In plastids, β-carotene is converted into zeaxanthin through the action of β-carotene hydroxylase (BCH). Zeaxanthine is converted into trans-violaxanthin by ZEP, encoded by *hp3*. Trans-violaxanthin is further modified into neoxanthin by NSY—encoded by *NDX1*—or into cis-violaxanthin by an enzyme still uncharacterized in tomato. Both intermediates are substrates for NCED—encoded by *NOT*—to be converted into xanthonin. Xanthonin moves across the plastid membrane to the cytosol for conversion into abscisic aldehyde first—a reaction catalyzed by SDR—and finally to ABA. This last step is catalyzed by the combined action of AAO and MoCo, respectively, encoded by *SIT* and *FLC*.

All the phenotypes observed in *not* mutant plants appear aggravated in the *flc* mutant (cv. Rheinlands Ruhm) obtained through x-ray mutagenesis. *flc* mutants wilt very quickly as a result of large and malfunctioning stomata, causing a rapid loss of water. Early dehydration is observed in the fruits of these mutants, which are smaller than the WT ones. Additionally, *flc* mutants have reduced plant height and leaf area despite their higher rates of gas exchange due to constitutively open stomatal conductance [23,47]. Strikingly, plants carrying both *not* and *flc* mutations synergically exaggerate all these phenotypes and, additionally, synthetize more ethylene in fruit pericarps—although this may be a pleiotropic effect due to the genetic cross between two different backgrounds [23,26]. The aggravated phenotype of *not flc* double mutants could depend on *not* and *flc* single mutants still accumulating significant levels of ABA, amounting to 47% and 21% of those of the WT plants, respectively [48]. Furthermore, *flc* mutants accumulate higher levels of proline and simple sugars—part and precursors of osmoprotective compounds, respectively—than WT controls, possibly in response to the more intense effects of water deprivation experienced by plant cells [27]. Therefore, ABA-deficient mutants can still elicit water deficit responses (albeit at lower levels), either through residual ABA production or the activation of pathways that lead to plant cell protection.

The *SIT* gene was identified as a homologue of *Arabidopsis thaliana* aldehyde oxydase 3 (*AtAAO3*) by Harrison and co-workers in 2011 [26] while studying the well-known wilting mutant *sit*. The growth retardation of the *sit* mutant is primarily attributable to compromised stomata function, which, like *not* and *flc*, is larger and constitutively opened, resulting in rapid water loss, faster wilting, and reduced WUE [24,49]. When comparing the *not*, *flc*, and *sit* mutants, the latter is the most affected in terms of leaf ABA deprivation [50]. Through heterografting experiments with WT, *sit*, and *flc* plants, it was discovered that ablating ABA biosynthesis site specificity has differential effects on plant responses to drought, salinity, and phosphate starvation. The effects of ABA depletion on leaf transpiration can be observed in *flc* and *sit* scions grafted onto WT rootstocks and not in WT scions with mutant rootstocks, implying shoot-synthetized ABA as the main driver of stomata closure [29,30].

## 2.2. Tomato ABA Signaling Mutants and Transgenic Plants

The core ABA signaling cascade is composed of three main proteins and has been excellently reviewed elsewhere [51–54]. Briefly, ABA is bound by a family of soluble receptors known as the pyrabactin resistance/pyr1-like/regulatory component of ABA receptors (PYR/PYL/RCAR, PYLs hereafter). Upon binding to ABA, PYLs interact with Clade A protein phosphatases 2Cs (PP2Cs). This interaction inhibits the phosphatase activity of PP2Cs, allowing their substrate, the protein kinases of the SNF1-related protein kinase 2 (SnRK2) group, to be phosphorylated. In the absence of ABA, PP2Cs bind SnRK2s and keep them in a dephosphorylated, inactive form. Activated SnRK2s can phosphorylate downstream components including basic leucine zippers (bZIPs)-type ABA-responsive transcription factors (TFs) called ABA responsive elements (ABREs)" binding proteins/abscisic acid responsive elements-binding factors (AREB/ABFs, ABFs hereafter). Phosphorylated ABFs enact the transcription of ABA/stress-response genes by directly binding to the ABRE sequences in their promoters. ABA signaling comprises other TFs of a wide variety of families, such as homobox proteins (HBs), MYB domain proteins (MYBs), NAC domain-containing proteins (NACs), basic helix–loop–helix (bHLH), and WRKY DNA-binding proteins (WRKYs). In tomato, the ABA signaling cascade is well conserved with Arabidopsis, although some confusion exists in its nomenclature (e.g., in [32], the PYL receptors are named differently than in subsequent works [55,56] and the two PP2Cs named *Sl*PP2C30 and *Sl*PP2C49 in [57] were previously referred to as *Sl*PP2C5 and *Sl*PP2C3 in [34] and [33], respectively). To avoid uncertainties, the nomenclature used hereafter is summarised in Table S1.

Experiments aimed at modifying ABA signaling genes have been described and an overview of the phenotypes resulting from this deregulation is provided in Figure 2.

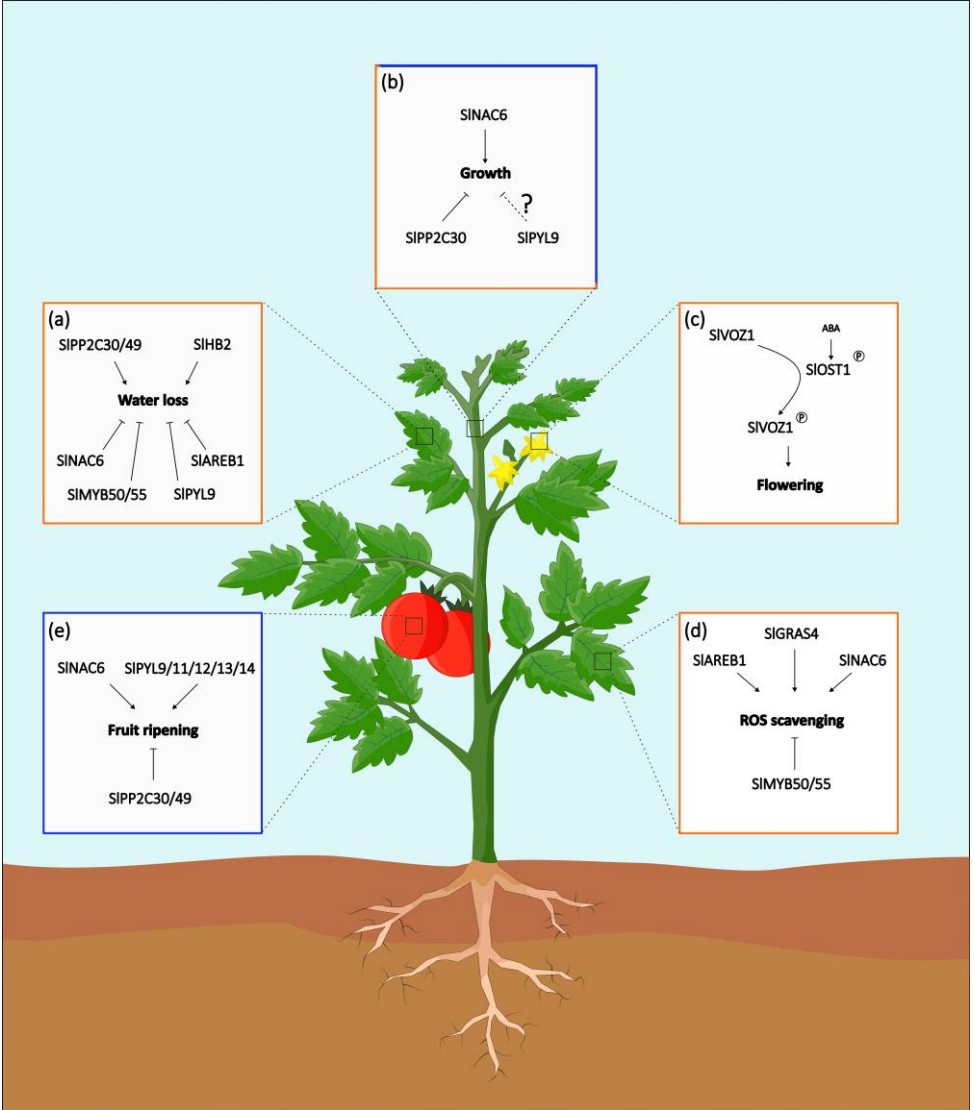

**Figure 2.** Phenotypic effects reported by manipulating ABA signaling components in tomato. (**a**) Leaf water loss is induced (arrow) by *Sl*PP2C30/49 and *Sl*HB2 and inhibited (blunted-end line) by *Sl*NAC6, *Sl*MYB50/55, *Sl*PYL9, and *Sl*AREB1. (**b**) Plant growth is inhibited by *Sl*PP2C30 and promoted by *Sl*NAC6, while *Sl*PYL9 effects are still unclear (dotted arrow). (**c**) ABA-activated *Sl*OST1 phosphorylates *Sl*VOZ1 to promote floral transition. (**d**) *Sl*AREB1, *Sl*GRAS4, and *Sl*NAC6 promote ROS scavenging activity, which is negatively regulated by *Sl*MYB50/55. (**e**) *Sl*PYL9/11/12/13/14 and *Sl*NAC6 stimulate fruit ripening while *Sl*PP2C30/50 inhibit it. Blue boxes represent phenotypes observed when plants are well-watered, orange boxes when they are under a water deficit, and orange/blue boxes under both conditions.

Fifteen PYL-coding genes were identified by Gonzalez-Guzman et al. [32], comprehensively defined with a common nomenclature in [55] and divided into three subfamilies. Tomato *SlPYL* transcripts are particularly abundant in roots. Some *SlPYLs* are also expressed in other vegetative and reproductive tissues: *SlPYL1/3/4/9/11*, which is consistently detected in leaves; *SlPYL2*, which is highly enriched in stems; *SlPYL7/13/14/15*, which is highly expressed at different fruit ripening stages; and *SlPYL5*, which is expressed in flowers [56]. The phenotypic effects of PYLs' modulation were studied mainly in fruit development. The main PYLs involved in fruit ripening are *SlPYL9/11/12/13/14*—even though their transcript can also be detected in other organs. The RNAi-mediated co-silencing of these four genes led to smaller, late-ripening and abnormally shaped fruits. Silenced plants also presented an increased number of fruit, but no overall benefit in terms

of yield [56]. Kai and colleagues [31] performed an in-depth characterization of *Sl*PYL9 in fruit ripening. *SlPYL9* expression is mainly observed in roots and leaves [32,56] but also occurs in ripening fruits (and in seeds), flower buds, flowers, and stems [31,56]. In developing fruits, *Sl*PYL9 promotes ABA accumulation and *SlPYL9-OE*-overexpressing plants displayed few or no defects in terms of fruit shape. The fruits were, however, quicker in their ripening process compared with the WT plants and their shelf-life was shortened. Conversely, *SlPYL9*-RNAi tomatoes were late-ripening and abnormally shaped, and displayed increased shelf-life. Besides these effects on fruit development, an important aspect is whether the modulation of *SlPYLs* could also affect tomato drought stress performances without compromising growth. As expected, *SlPYL9-RNAi* plants increased water loss under water deficits whereas the opposite occurred in *SlPYL9-OE* plants, confirming the role of this receptor in ABA-mediated drought tolerance [31]. However, both *SlPYL9* overexpression and silencing led to a reduced size overall, indicating that alterations of ABA signaling through *SlPYL9* may generally impact plant growth. Recently, a VIGS approach was used to transiently silence *SlPYL4* [58]. When subjected to osmotic stress, *SlPYL4* plants showed lower overall ROS scavenging performances and faster wilting compared to controls, although a direct comparison with *SlPYL9* RNAi plants would be informative to define gene-specific effects under the same experimental conditions.

Qiu and collaborators [57] identified 92 *PP2Cs* (*SlPP2C1-92*) divided in eight subfamilies (A-H), among which 70 present ABRE in their promoter region; of these, only *Sl*PP2C28, *Sl*PP2C30, *Sl*PP2C39, *Sl*PP2C49, and *Sl*PP2C55 were demonstrated to interact with PYR/PYL/RCAR receptors and SnRK2s in a yeast two-hybrid assay [31,59]. Changes in *SlPP2C49* and *SlPP2C30* transcript levels had effects on both fruit ripening/development and plant resilience to water deprivation [33,34]. Suppressing *SlPP2C49* and *SlPP2C30* expression led to reduced leaf water loss under water deficits and quicker fruit ripening, while their overexpression increased sensitivity to water deprivation and delayed fruit maturation. A reduction in plant growth was observed only in *SlPP2C49-OE* transgenic plants, suggesting that the two PP2Csfunctions are not completely redundant [33,34]. In Arabidopsis, *SlPP2C49* homologues *AtHAI1/2/3* primarily regulate dehydrin and aquaporin coding genes and the accumulation of osmoregulatory compounds (e.g, proline) [60], with a minor role in the control of stomatal conductance [61]. Thus, more work is needed to identify the specific molecular roles of PP2Cs in tomato drought stress responses.

A total of nine SnRK2s are putative members of the ABA signaling cascade in tomato: *Sl*SnRK2.1 to *Sl*SnRK2.8, including *Sl*SnRK2.3 which has been renamed as *S. lycopersicum* open stomata 1 (*Sl*OST1) [15,34]. These proteins were shown to interact with PYLs through Y2H assays [59] and the formation of SnRK2-ABFs complexes was observed for *Sl*SnRK2.5 in yeast [59] and *Sl*SnRK2.4 in both yeast and *Nicotiana benthamiana* [62]. In Arabidopsis, *OST1* is a key regulator of guard cell activity. Mutants of *ost1* are hyposensitive to ABA and osmotic stress and highly drought susceptible [63]. *Sl*OST1 is functionally conserved in regulating guard cell movements in tomato. The CRISPR-Cas9-mediated mutagenesis of *SlOST1* in cv. microtom and Alisa Craig caused stomata hyposensitivity to drought and osmotic stress [15]. Interestingly, *Slost1* mutants display a late flowering phenotype under both normal and water-deficit conditions, independent of the photoperiod. These effects on flowering are mediated by the transcription factor vascular plant one zine finger 1 (*Sl*VOZ1), which is phosphorylated by *Sl*OST1 upon ABA perception. The phosphorylation of VOZ1 promotes its stabilization and nuclear accumulation, leading to binding at the promoter of *SINGLE FLOWER TRUSS* (*SFT*)—the main florigen gene—enhancing its expression in leaves [15]. Interestingly, *not* and *flc* mutants show an extended vegetative phase compared with WT plants, which might support a positive role for ABA in promoting the transition between vegetative and reproductive growth in tomato [64]. The flowering time is a major driver of productivity and agronomic yield [65]. The above results indicate that alterations in ABA signaling in tomato may also alter the flowering time through changes in *SFT* accumulation. Thus, uncoupling flowering time from ABA signaling would allow us to

obtain improved water-deficit responses without compromising the yield gains obtained through the optimization of the flowering time [66].

While changes in the early ABA signaling processes may lead to pleiotropic effects on plant growth, altering downstream ABA-regulated TFs could lead to more spatially and temporally fine-tuned ABA sensitivity. The tomato bZIP TFs family consists of 69 bZIPs named bZIP1-69. These are further divided into 11 groups, whereby group VI includes 11 homologues of the Arabidopsis ABFs: *Sl*bZIP02, *Sl*bZIP08, *Sl*bZIP09, *Sl*bZIP14, *Sl*bZIP31, *Sl*bZIP33/*Sl*AREB1, *Sl*bZIP52, *Sl*bZIP54, *Sl*bZIP56, *Sl*bZIP61, and *Sl*bZIP65/*Sl*AREB2 [67]. Evidence for interactions with SnRK2s has been provided for *Sl*bZIP33/*Sl*AREB1 and *Sl*bZIP65/*Sl*AREB2 [59,62]. *SlAREB1* is also functionally characterized in tomato [35,36]. *SlAREB1*-silenced tomatoes under a prolonged severe drought wilt earlier than WT controls. Conversely, the overexpression of *SlAREB1* conferred drought resistance to the plants, which wilted two weeks afterwards and performed better in terms of photosynthetic efficiency and ROS balance compared with the WT plants. No significant effects on growth were observed as a result of the *SlAREB1* overexpression or silencing [35,36] except for a slight decrease in the shoot length [36].

The genetic manipulation of different types of ABA-regulated TFs has been shown to produce clear deregulation in stomata function. *SlNAC6* overexpression reduces water loss by exaggerating stomatal sensitivity to ABA and increasing the ROS scavenging performances of plant cells. Conversely, its downregulation results in the opposite phenotype. However, the manipulation of *SlNAC6* also led to pleiotropic developmental defects as *SlNAC6-RNAi* plants are dwarf plants, whereas *SlNAC6-OE* tomatoes are late-flowering under standard growth conditions and are characterised by faster fruit-ripening [37]. The overexpression of the ABA-induced *SlbHLH69* confers enhanced stomatal closure specifically in response to osmotic stress without altering plant development—although some agriculturally important traits, such as flowering time, fruit development, and yield, were not assessed [42].

Several MYB-type TFs are also part of the tomato ABA signaling cascade, in which they usually act as negative regulators of its signaling. The silencing of *SlMYB50* and *SlMYB55* caused increased drought and salinity tolerance [39,40]. Under a water deficit, these transgenic lines show reduced leaf water loss and are less prone to chlorosis, with better ROS detoxification performances compared with those of WT plants. *SlMYB55*-RNAi plants also flowered earlier and produced more flowers than the WT plants under irrigated conditions [40], although whether this phenotype was due to increased ABA accumulation/signaling warrants further investigation. Indeed, other ABA-regulated TFs were shown to regulate different ABA signaling components. For instance, *S. lycopersicum* gibberellic acid insensitive (GAI) repressor of *ga1-3* (RGA) scarecrow (SCR) 4 (*SlGRAS4*) was shown to interact with *Sl*AREB1, *Sl*AREB2, and *Sl*Snrk2.4, to transcriptionally regulate several ABA signaling-encoding genes [38]. Enhancing *SlGRAS4* levels led to improved ROS detoxification during a prolonged water deficit and prevented leaf chlorosis. Another ABA-regulated TF, *Sl*HB2, was shown to negatively regulate several stress-responsive genes. The silencing of *SlHB2* improved water loss and proline accumulation while reducing the oxidative damage in high-salinity and water-deficit conditions without altering plant development [41]. Thus, because TFs play a key role in determining the unique characteristics and functions of different cell types, they may represent primary targets to manipulate ABA responses at the tissue level.

## 3. Conclusions and Future Perspectives

The evidence reviewed here suggests that we are only beginning to address how changes in ABA signaling and production can be exploited to confer superior productivity under water-deficit conditions in tomato, and whether this approach can be harnessed without causing negative effects on plants' different life history traits, growth, and fruit maturation. This could be challenging as changes in ABA signaling or accumulation could impact, for example, hormonal balance, including GA signaling. The relation between

ABA and GA in guard cells in tomato has been well studied [68,69]. The activation of GA signaling in seeds suppresses desiccation tolerance due to the inhibition of ABA-induced gene expression [70], indicating that GA and ABA negatively interact at the hormone signaling (and also biosynthesis) levels in many plant organs and tissues. Variations in ABA accumulation and signaling are linked to multiple pathways controlling plant growth and different aspects of reproductive development, including floral transitions and fruit/seed development/maturation. To circumvent these limitations, one strategy would be to seek approaches to rewire cell/organ-specific ABA regulation, although prior knowledge about the expression of ABA regulators at a high spatial/temporal resolution would be critical. Gene-editing techniques could subsequently be used to generate mutants in ABA-related genes expressed in the target domains [71]. Instead of targeting coding sequences, the same approach could be used to mutate cis-regulatory sequences of ABA-related genes. Targeting cis-regulatory elements located in the promoters of tomato domestication genes has proven to be successful in reprogramming transcriptional regulation and generating novel phenotypic variation [72]. Because cis-regulatory elements can be organized as independent modules, different mutations in the promoter regions could influence different aspects of gene regulation, e.g., inducibility by water deficits or type of cell/tissue expression. Indeed, examples of modular chimeric promoters to engineer guard cell activity have been previously described [73]. The above strategies could be further adapted to isolate weak alleles of target ABA-related genes of interest. A weak allele is a variant of a gene that has reduced activity or function compared to WT plants. This can occur due to mutations in the DNA sequence of the gene or changes in the regulation of gene expression. Weak alleles can result in mild or subtle phenotypic effects and may be less likely to cause deleterious pleiotropic effects under optimal growth conditions [74].

In summary, increasing our knowledge about the patterns of expression of specific ABA components and their interactors may enable us to engineer strategies to separate undesirable ABA effects on developmental/growth traits from enhanced drought resistance and yield stability traits.

**Supplementary Materials:** The following supporting information can be downloaded at: https://www.mdpi.com/article/10.3390/ijpb14010014/s1, Table S1: ABA signalling genes names used in this paper.

**Author Contributions:** Conceptualization, P.K.K. and S.C.; Writing—Original Draft Preparation, P.K.K., S.C. and A.S.; Writing—Review and Editing: L.C., P.K.K. and D.M.; Supervision, L.C. All authors have read and agreed to the published version of the manuscript.

**Funding:** Our research in the Conti lab was supported by a Research Grant from HFSP Ref.-No: RGP0011/2019 by the Italian Ministry of Agriculture (MiPAAF), project BIOTECH-Cisget. D.M. is supported by a research fellowship co-funded by the European Union—FSE, PON Ricerca e Innovazione 2014–2020. This study was carried out within the Agritech National Research Center and received funding from the European Union Next-GenerationEU (PIANO NAZIONALE DI RIPRESA E RESILIENZA (PNRR)—MISSIONE 4 COMPONENTE 2, INVESTIMENTO 1.4—D.D. 1032 17/06/2022, CN00000022). This manuscript reflects only the authors' views and opinions, neither the European Union nor the European Commission can be considered responsible for them.

**Conflicts of Interest:** The authors declare no conflict of interest.

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
