# Peer review of "How Changes in ABA Accumulation and Signaling Influence Tomato Drought Responses and Reproductive Development"

_2037-0164, doi:10.3390/ijpb14010014_

Round 1

Reviewer 1 Report

The manuscript entitled “How changes in ABA accumulation and signalling influence tomato drought responses and reproductive development” by Krukowski et al., reviewed the accumulated knowledge on ABA in tomato comprehensively. 

Molecular genetic studied using Arabidopsis has successfully unveiled the mechanisms that are employed in the ABA synthesis, ABA signaling and expression of ABA responsive genes. I think the reviewing on ABA related literature of tomato as a crop should emphasize its uniqueness or differences from Arabidopsis. It would be very interesting if the author could offer many ideas how we can improve drought tolerance of tomato or other crops by modulating ABA related components and mentioned in the abstract. However, I am not sure the author successfully offered them enough.  And I also felt uneasiness when reading the conclusions. It described mainly about GA, which is not mentioned in the main text. I think the author have the conclusion of tomato ABA research and the possibility of improving the crop using the obtained knowledge besides addressing GA. Moreover, I found that major parts of the text tended to describe the literatures enumeratively without any comments that would help understanding. Summarizing accumulated literatures would be useful and helpful to capture the status of this research field…, but I hope the author can improve this manuscript to more thought-provoking one.

I also concern about the quality of the text. I think the latter part (after line 202) should be improved. I found many improper expressions. For example, 

I never saw the expression such as “by [32]”, “[56] recognized…” (line 237, 265). 

Line 245-247, this sentence can be concise.

Line 262, (Li et al, 2022), and “When subject-ed to…”

Line 271, references 33 and 34 are collect? need check, I think.

Line 268-270, what is the point of this sentence?

Line 309-320, “conversely” appeared three times in a paragraph. 

Figure2, SlPYL8,9,10, 11, 12 are shown to be involved in fruit ripening. But in the main text, SlPYL9, 11, 12, 13, 14 were mentioned to be, and there was no description about SlPYL8 in the main text.

so on…

And the former half has typos (for example, L180, Under water”,” deficit).

I think the author should edit the text more carefully again or consult with a professional English editor. 

Finally, I have several scientific questions.

Line 161, the not is null mutation, while only VIGS silenced SlNCED1 fruit ripening delay. How this happen. I think the author should give some idea to understand this discrepancy. Can VIGS system always affect fruit ripening? 

Line 184 (and Figure 1), why null mutation in the sole NCED gene in tomato (is it right?) cannot remove ABA completely? Any explanations? Is there any chance tomato has another NCED gene? 

Line 284, “OST1 is a guard cell-specific kinase.” What does this mean? OST1 is not “stomata specific”. 

Author Response

Reviewer 1

The manuscript entitled “How changes in ABA accumulation and signalling influence tomato drought responses and reproductive development” by Krukowski et al., reviewed the accumulated knowledge on ABA in tomato comprehensively. 

Molecular genetic studied using Arabidopsis has successfully unveiled the mechanisms that are employed in the ABA synthesis, ABA signaling and expression of ABA responsive genes. I think the reviewing on ABA related literature of tomato as a crop should emphasize its uniqueness or differences from Arabidopsis. It would be very interesting if the author could offer many ideas how we can improve drought tolerance of tomato or other crops by modulating ABA related components and mentioned in the abstract. However, I am not sure the author successfully offered them enough.  And I also felt uneasiness when reading the conclusions. It described mainly about GA, which is not mentioned in the main text. I think the author have the conclusion of tomato ABA research and the possibility of improving the crop using the obtained knowledge besides addressing GA. Moreover, I found that major parts of the text tended to describe the literatures enumeratively without any comments that would help understanding. Summarizing accumulated literatures would be useful and helpful to capture the status of this research field…, but I hope the author can improve this manuscript to more thought-provoking one.

Thank you for your valuable comments and we are sorry our manuscript did not fully meet your expectations.

Our goal here was to present an up-to-date summary of the genetic and transgenic resources available in tomato to modulate ABA signalling and whether these could be (or were) used to obtain improvements in plant performances under water deficit conditions. This choice was mainly dictated by reports from wheat and Arabidopsis suggesting that changes in ABA sensitivity could ameliorate drought tolerance traits without growth penalties.

We had to navigate through a wide literature (characterized by ambiguous nomenclature – e.g. see Supporting table) often describing organ-specific drought responses under non-standardized conditions.  We thus felt that organizing this body of references following the logic of the Arabidopsis ABA signalling structure would be easier to read. However, we agree that we may have missed some of the “crop-specific” angles. All in all, we did our best to be as comprehensive as possible, and to emphasize the gaps in our knowledge about the genetic perturbation of ABA signalling in tomato. In fact, very few genes have been perturbed through overexpression or silencing and highly pleiotropic phenotypes were observed. In this context, the reference to GA serves as an example of tightly linked processes that would be hard to disentangle without prior knowledge about patterns of gene expression and action.

Following the reviewer’s suggestions, we expanded and clarified these points in the concluding section (now renamed conclusions and future perspective), where we identified potential strategies to break the above-mentioned pleiotropy using the Crispr-Cas9 technology, although suitable target genes remain to be identified.

I also concern about the quality of the text. I think the latter part (after line 202) should be improved. I found many improper expressions. For example, 

I never saw the expression such as “by [32]”, “[56] recognized…” (line 237, 265). 

We apologize for these issues. We carefully re-checked all the manuscript to improve the citation style.

Line 245-247, this sentence can be concise.

We have improved this sentence

Line 262, (Li et al, 2022), and “When subject-ed to…”

Thanks, we have corrected this

Line 271, references 33 and 34 are collect? need check, I think.

We double checked and the references are correct. Differences in gene names between our manuscript and cited papers are discussed and summarized in table S1.

Line 268-270, what is the point of this sentence?

The reviewer is correct. We have now removed this sentence as the main idea was captured already in the previous sentence.

Line 309-320, “conversely” appeared three times in a paragraph. 

Thank you for pointing this out. We have corrected this inproper repetition.

Figure2, SlPYL8,9,10, 11, 12 are shown to be involved in fruit ripening. But in the main text, SlPYL9, 11, 12, 13, 14 were mentioned to be, and there was no description about SlPYL8 in the main text.

Thank you for pointing this out. We have corrected the figure and the legend consititent to the text (i.e. SlPYL9/11/12/13/14  being the main genes involved in fruit ripening) 

so on…

And the former half has typos (for example, L180, Under water”,” deficit).

I think the author should edit the text more carefully again or consult with a professional English editor. 

We highly appreciate the reviewers’ helpful comments on this. We have carefully rewritten or reorganized several sentences to enhance readability and make some statements clearer. In general, the manuscript has been improved also in terms of quality of English language and clarity.

Finally, I have several scientific questions.

Line 161, the not is null mutation, while only VIGS silenced SlNCED1 fruit ripening delay. How this happen. I think the author should give some idea to understand this discrepancy. Can VIGS system always affect fruit ripening? 

Thanks for pointing this out. No information regarding fruit ripening of not is provided in literature. As far as we know, no effect of VIGS on tomato fruit ripening was ever reported. We changed the word “observed” with “assessed” to clarify this point.  

Line 184 (and Figure 1), why null mutation in the sole NCED gene in tomato (is it right?) cannot remove ABA completely? Any explanations? Is there any chance tomato has another NCED gene? 

Thanks, this information was incorrect. In the revised manuscript we highlight the presence of the SlNCED2 gene. It is thus possible that NCED2 activity compensates for loss in NCED1.

Line 284, “OST1 is a guard cell-specific kinase.” What does this mean? OST1 is not “stomata specific”. 

We agree. We have now clarified this by changing in “OST1 is a key regulator of guard cells activity”

Reviewer 2 Report

Good summary article on the role of ABA in tomato drought response and reproductive development.

Article with good and numerous references in literature.

Interesting conclusions from a practical point of view but to be further explored for a greater knowledge of the pathways involved that can be exploited for effective strategies in agriculture.

Figures 1 and 2 are missing, without which it is more difficult to understand the article.

They wrote 2 times paragraph 2.1

Author Response

Good summary article on the role of ABA in tomato drought response and reproductive development.

Article with good and numerous references in literature.

Interesting conclusions from a practical point of view but to be further explored for a greater knowledge of the pathways involved that can be exploited for effective strategies in agriculture.

Thank you for your positive comments on our study. In the revised version we have carefully revised the text and enhanced overall clarity and correctness. 

Figures 1 and 2 are missing, without which it is more difficult to understand the article.

Figures should now be correctly embedded in the revised version

They wrote 2 times paragraph 2.1

Thanks for pointing out this typo, we have now corrected the paragraph heading